# A Retrospective Study of Complications of Enteral Feeding in Critically Ill Children on Noninvasive Ventilation

**DOI:** 10.3390/nu15122817

**Published:** 2023-06-20

**Authors:** Montserrat Sierra-Colomina, Nagam Anna Yehia, Farhan Mahmood, Christopher Parshuram, Haifa Mtaweh

**Affiliations:** 1Department of Pediatric Critical Care, Toulouse Children’s University Hospital, 31300 Toulouse, France; sierracolomina.m@chu-toulouse.fr; 2Department of Nutritional Sciences, University of Toronto, Toronto, ON M5S 1A8, Canada; anna.yehia@mail.utoronto.ca; 3Faculty of Medicine, University of Ottawa, Ottawa, ON K1H 8M5, Canada; fmahm059@uottawa.ca; 4Department of Critical Care Medicine, The Hospital for Sick Children, Toronto, ON M5G 1X8, Canada; chris@sickkids.ca

**Keywords:** enteral nutrition, noninvasive ventilation, respiratory failure, critical care, pediatric

## Abstract

The utilization of noninvasive ventilation (NIV) in pediatric intensive care units (PICUs), to support children with respiratory failure and avoid endotracheal intubation, has increased. Current guidelines recommend initiating enteral nutrition (EN) within the first 24–48 h post admission. This practice remains variable among PICUs due to perceptions of a lack of safety data and the potential increase in respiratory and gastric complications. The objective of this retrospective study was to evaluate the association between EN and development of extraintestinal complications in children 0–18 years of age on NIV for acute respiratory failure. Of 332 patients supported with NIV, 249 (75%) were enterally fed within the first 48 h of admission. Respiratory complications occurred in 132 (40%) of the total cohort and predominantly in non-enterally fed patients (60/83, 72% vs. 72/249, 29%; *p* < 0.01), and they occurred earlier during ICU admission (0 vs. 2 days; *p* < 0.01). The majority of complications were changes in the fraction of inspired oxygen (220/290, 76%). In the multivariate evaluation, children on bilevel positive airway pressure (BiPAP) (23/132, 17% vs. 96/200, 48%; odds ratio [OR] = 5.3; *p* < 0.01), receiving a higher fraction of inspired oxygen (FiO_2_) (0.42 vs. 0.35; *OR* = 6; *p* = 0.03), and with lower oxygen saturation (SpO_2_) (91% vs. 97%; *OR* = 0.8; *p* < 0.01) were more likely to develop a complication. Time to discharge from the intensive care unit (ICU) was longer for patients with complications (11 vs. 3 days; *OR* = 1.12; *p* < 0.01). The large majority of patients requiring NIV can be enterally fed without an increase in respiratory complications after an initial period of ICU stabilization.

## 1. Introduction

Acute lower respiratory tract infections are the leading cause of hospital and intensive care unit (ICU) admissions in the pediatric population [1]. In cases progressing to acute respiratory failure, noninvasive ventilation (NIV) support is increasingly being utilized to improve oxygenation, ventilation, and work of breathing [2,3,4,5,6,7]. Although NIV use has been associated with a decreased need for intubation, it has also been described as a common risk factor for both delayed enteral nutrition (EN) initiation and underfeeding in the pediatric intensive care unit (PICU) [8,9,10,11]. Nutritional guidelines recommend initiating EN within the first 24–48 h of admission to ICU to ensure adequate nutrition, preserve gastric mucosal integrity, and maintain motility [12,13]. Nevertheless, poor nutritional intake is present in up to one-third of hospitalized patients within the United States of America and Europe [10,11,14,15]. This is associated with longer ICU and hospital length of stay, higher risk of hospital acquired infections, and increased mortality [11,12,16,17,18]. The apparent reluctance to provide nutritional support during NIV has been attributed to multiple causes including concerns of secondary gastric distension affecting diaphragmatic function and compromising respiration, variations in gastric pressure contributing to reflux and bronchial aspiration, the increased risk of developing swallowing disorders [8,19,20], a poor NIV mask seal due to the presence of a nasogastric tube [21], and the concern for respiratory deterioration that would require intubation and an empty stomach to reduce procedural complications [9,11,15,22,23,24,25].

Although data are limited, the available evidence suggests variability in the incidence of respiratory (0–12%) [10,11,22] and gastric complications (4.8–20%) [3,23] when providing EN to pediatric patients on NIV [26]. The objective of this study was to evaluate the safety of enteral feeds in children requiring NIV for acute respiratory failure. Safety was defined by the absence of extraintestinal complications.

## 2. Materials and Methods

We performed a single-center retrospective cohort study at the Hospital for Sick Children (SickKids), Toronto, Canada (41 beds). Eligible children were less than 18 years of age, admitted to PICU over a 5 year span (2012–2017), and had a respiratory illness treated by NIV. NIV was defined as continuous positive airway pressure (CPAP) or bilevel positive airway pressure (BiPAP). We excluded children managed with humidified high-flow nasal cannula (HHFNC), those requiring home oxygen or home NIV, and those with congestive heart failure, abdominal sepsis, enterocolitis, or perforation of the gastrointestinal tract. Ethics approval was obtained from the SickKids research ethics board.

The primary outcome was the association between respiratory complications and provision of enteral nutrition in children on NIV. The complications included an increase in oxygen requirement, defined as a significant change in fraction of inspired oxygen (FiO_2_) from baseline (Table 1) sustained for two or more hours, increase in pressures defined as an increase of 4 cmH_2_O or more in one of the pressures on NIV, intubation, apnea defined as an episode of lack of respiratory effort that required the application of bag-mask ventilation, or cardiorespiratory arrest. Enteral feeding was defined as the delivery of nutrients into the gastrointestinal tract for more than 4 h while on NIV. Secondary outcomes were ICU and hospital length of stay (LOS).

Patient descriptors and primary ICU diagnosis at the time of ICU admission were recorded. The following data were abstracted hourly or as available in the chart from time of initiation of NIV until NIV discontinuation or ICU discharge: vital signs including respiratory rate (RR), heart rate (HR), systolic blood pressure (SBP), diastolic blood pressure (DBP), and oxygen saturation (SpO_2_); respiratory support including FiO_2_ and pressures on CPAP and BiPAP; enteral feeds including hourly volume standardized to weight and enteral medication volumes and flushes. For patients who developed a complication, we recorded vital signs and enteral feed rate around the time of complication as available in the chart. For those without a complication the median vital signs and maximum enteral nutrition volume received while on NIV was utilized for comparison. 

We described baseline patient characteristics for the total cohort by age subgroups (<1 year, 1–5 years, 5–10 years, and >10 years) using measures of central tendency and spread to describe continuous variables and frequency and percentages for categorical variables. We compared baseline patient characteristics, vital signs, NIV settings, length of stay, and mortality for those who received enteral feeds vs. those who were not fed using *t*-test or Mann–Whitney according to the distribution of continuous variables, or the chi-square/Fisher exact test for categorical variables. In patients who developed a complication, we compared patient characteristics for those who received enteral feeds vs. those not fed. We compared patients’ vital signs at the time of decision to initiate feeds and prior to developing a complication using a paired-samples *t*-test. To determine the association of patient factors and development of complications, we used logistic regression and evaluated the first complication each patient developed against factors of interest. Then, we used a generalized estimating equation to evaluate the association of enteral feeds and vital signs with the development of any complication during the admission, thus accounting for repeated measures. Initial univariate evaluations were performed, and factors with *p* < 0.1 in univariate models were then included in a multivariate model. Factors with *p* < 0.05 in the multivariate model were defined as significantly associated with complications. Assumption testing for the multivariate model was performed, and results were presented as odds ratio with 95% confidence interval. We generated Kaplan–Meier curves to estimate the probability of experiencing a complication, and we conducted the log-rank test to assess the significance of differences in complication-free survival curves between groups. Lastly, we matched cases from our cohort on the basis of age, weight, and NIV settings to calculate the risk ratio of a respiratory complication while being enterally fed. All analyses were performed in Statistical Package for Social Science Software Version 19.0 (IBM Corporation, Armonk, NY, USA).

## 3. Results

### 3.1. Demographics

We identified a total of 555 patients requiring NIV for respiratory illness during a period of 5 years and included 332 who met the inclusion criteria (Figure 1). The mean ± standard deviation (M ± SD) age was 4.85 ± 5.57 years, and 204 (61.4%) were male. The mean ICU and hospital length of stay were 6.27 ± 9.36 days and 25.66 ± 38.1 days, respectively. The majority of patients (N = 249, 75%) were enterally fed while on NIV (Table 2). Fourteen percent of the cohort received mechanical ventilation at some point during their PICU admission, and 21 (6%) patients died in hospital.

### 3.2. Description of Vital Signs and Noninvasive Respiratory Support

At the time of EN initiation, the entire patient cohort had a median (first quartile, third quartile) (MD [Q1, Q3]), HR of 134 (113, 152) beats/min, SBP of 101 (91, 112) mmHg, DBP of 58 (48, 70) mmHg, RR of 34 (25, 44) breaths/min, SpO_2_ of 98% (95%, 99%), and FiO_2_ of 0.3 (0.2, 0.4). BiPAP support was utilized in 206/332 (62%) and CPAP was utilized in 126/332 (38%) patients. Table 2 summarizes patient characteristics according to age cohort. Among children requiring BiPAP at the time of EN initiation, the median (Q1, Q3) inspiratory positive airway pressure (IPAP) was 14 (12, 15) cmH_2_O and the median expiratory positive airway pressure (EPAP) was 7 (6, 8) cmH_2_O. For those on CPAP at EN initiation, the median pressure was 7 (6, 8) cmH_2_O.

**Table 2 nutrients-15-02817-t002:** Patient characteristics by age group (N = 332).

Characteristics	Statistic	<1 Year	1–5 Years	5–10 Years	>10 Years
Number of patients	N (%)	130 (39)	83 (25)	46 (14)	73 (22)
Sex, male	N (%)	85 (65)	56 (68)	29 (63)	34 (47)
Weight, kg	MD (Q1, Q3)	4.6 (3.6, 5.9)	11 (9.4, 13.9)	22.2 (18, 25.9)	43 (32, 55)
NIV settings, cmH2O					
CPAP	MD (Q1, Q3)	7 (6, 7)	6 (6, 8)	7 (6, 8)	7 (6, 10)
BiPAP (EPAP)	MD (Q1, Q3)	7 (6, 7)	7 (6, 8)	7 (6, 8)	7 (6, 8)
BiPAP (IPAP)	MD (Q1, Q3)	14 (12, 15)	14 (12, 16)	14 (12, 14)	14 (12, 16)
FiO_2_	MD (Q1, Q3)	0.3 (0.25, 0.4)	0.35 (0.25, 0.6)	0.35 (0.3, 0.45)	0.35 (0.3, 0.5)
Vital signs					
Heart rate	MD (Q1, Q3)	143 (131, 161)	132 (114, 149)	123 (104, 140)	115 (93, 138)
Respiratory rate	MD (Q1, Q3)	36 (30, 52)	34 (25, 41)	35 (25, 41)	24 (20, 30)
SBP, mmHg	MD (Q1, Q3)	94 (85, 107)	105 (94, 115)	104 (95, 114)	106 (97, 117)
DBP, mmHg	MD (Q1, Q3)	55 (47, 67)	59 (48, 71)	62 (51, 71)	63 (50, 71)
Oxygen saturation, %	MD (Q1, Q3)	98 (96, 100)	98 (95, 100)	97 (95, 98)	97 (95, 99)
On enteral feeds	N (%)	96 (74)	65 (78)	37 (80)	51 (70)
Complication	N (%)	56 (43.1)	31 (37.3)	14 (30.4)	31 (42.5)
PICU LOS, days	MD (Q1, Q3)	5 (2, 9)	3 (1, 6)	2 (1, 4)	2 (1, 7)
Hospital LOS, days	MD (Q1, Q3)	15 (6, 44)	8 (4, 24)	10 (6, 22)	11 (7, 18)
Deaths	N (%)	5 (4)	8 (10)	3 (6)	5 (7)

The cohort included a wide age range representative of the PICU population. The largest group involved children under 1 year of age. BiPAP was utilized more frequently than CPAP, and the whole cohort had similar NIV settings. The incidence of respiratory complications was similar across the different age groups. Most patients received enteral nutrition (75%), and feeding rates were higher in younger children. A longer PICU LOS was observed in younger children. N, number; MD, median; Q1, first quartile; Q3, third quartile; NIV, noninvasive ventilation; CPAP, continuous positive airway pressure; BiPAP, bilevel positive airway pressure; EPAP, expiratory positive airway pressure; IPAP, inspiratory positive airway pressure; FiO_2_, fraction of inspired oxygen; SBP, systolic blood pressure; DBP, diastolic blood pressure; PICU, pediatric intensive care unit; LOS, length of stay.

The baseline characteristics, NIV settings, and vital signs while on NIV were similar between patients who received enteral feeds and those who did not (Table 3).

Patients who developed a complication had similar respiratory rate and blood pressure at the time of decision to initiate enteral nutrition and the time of development of the complication (Table 4). However, there was a statistical difference detected in heart rate and oxygen saturations between the two time periods.

### 3.3. Complications 

There were 132 patients (132/332, 40%) who had 290 complications, with a change in FiO_2_ being the most frequent (Figure 2). Of the 132 patients, 72 (54%) had one complication and 60 (45%) patients had more than one complication. In those with more than one complication, the most frequent first complication was an increase in FiO_2_ (36/60, 60%), followed by intubation (21/60, 35%). Almost half of the patients with recurrent complications experienced only repeated increases in FiO_2_ (28/60, 47%). In seven of the 60 patients, the initial change in FiO_2_ was followed by a care escalation, while, in 53 of the 60 patients experiencing repeated complications, the first complication was the worst they developed. Patients who experienced multiple complications were not at higher risk of developing apnea, being intubated, or having a cardiorespiratory arrest. 

The first complication on NIV occurred earlier during admission in children not enterally fed at a median (Q1, Q3) of 0 (0, 3) days versus 2 (0, 4) days (*p* < 0.01) in those enterally fed. Patients who were nil per os (NPO) were significantly more likely to develop earlier a complication other than a change in FiO_2_ (log rank 10.56; *p* < 0.01) (Figure 3). A higher number of complications was noted in patients not fed enterally (60/83, 72.3%) in comparison to those receiving enteral nutrition (72/249, 28.9%; *p* < 0.01). The risk ratio for development of complication for those enterally fed was 0.4.

### 3.4. Factors’ Relationship with First Complication

In patients who developed a complication, there was no significant difference in age, sex, NIV settings, vital signs, length of stay, or mortality between those who were enterally fed or not (Table 5).

The factors significantly associated with the development of a first complication in the univariate assessment were being on BiPAP for respiratory support, having higher FiO_2_, having a lower blood pressure and oxygen saturation, not being on enteral feeds, or having lower feeding volumes for those fed. The first complication was also associated with increased PICU and hospital length of stay and mortality (*p* < 0.01). The multivariate logistic regression demonstrated an association between developing a first complication and being on BiPAP for respiratory support and requiring a higher FiO_2_. The odds of development of a complication were 5.3 times for those on BiPAP (Table 6). Time to discharge from ICU was longer for patients with complications (11 ± 13 vs. 3 ± 4 days; *OR* = 1.12; *p* < 0.01).

### 3.5. Factors’ Relationship with Any Complication

The development of a complication, including repeated complications, was associated with not being on enteral feeds, and having lower feeding volumes when on enteral nutrition, a higher heart rate, and a lower oxygen saturation in the univariate assessment. The generalized estimating equation demonstrated an association between developing a complication and lack of enteral nutrition and lower oxygen saturation. The odds of development of a complication for those not enterally fed was 3.6 times those who were enterally fed (Table 7).

We compared 83 enterally fed patients with fasting patients matched for age, weight, and NIV settings. There was no difference in their baseline vital signs (*p* > 0.05). The rate of complications was significantly lower in enterally fed patients (25/83, 30.1% vs. 60/83, 72.3%; *p* < 0.01) matched for age, weight, and NIV settings, with a risk ratio of 0.42 for enterally fed patients, similar to the entire cohort. 

## 4. Discussion

The incidence of respiratory and gastric complications in pediatric patients enterally fed while on NIV is variable [10,11,22,26]. Such differences may arise from the important variation in the definition of complication, the population studied, and the variable clinical practice between centers [3,10,15,22,23,24]. Trials comparing different NIV interfaces, EN timing, and route (pre- vs. post-pyloric feed) or type (intermittent vs. continuous) of EN have not shown a reduction in complications or an increase in the number of patients receiving adequate enteral nutrition while on NIV [21,22,27]. Our study evaluated the types of extraintestinal complications in critically ill children on noninvasive ventilation and the risk of developing a complication while receiving enteral feeds. The main finding was that enteral nutrition is not associated with increased rates of complications while on noninvasive ventilation for acute respiratory failure.

Firstly, we demonstrated in this study that the development of a complication on NIV was not associated with enteral nutrition delivery or the volume delivered in the 1 h or 4 h prior to development of a complication. Patients who were NPO or receiving a lower rate of enteral feed volume had significantly more respiratory complications and their complications occurred earlier during their admission. A potential hypothesis for this finding could be that clinicians identified patients with concerning respiratory presentations at high risk of potential deterioration and, thus, withheld nutrition. However, the vital signs and respiratory settings presented in this study do not support this hypothesis. The findings here are similar to those described by Tume and colleagues in a cohort of children requiring NIV, predominantly HHFNC, where respiratory complications occurred in 12.3% of children and were mainly non-severe, with 1.5% of patients suffering from a pulmonary aspiration [24]. Another recent study including asthmatic patients on BiPAP showed no complications secondary to EN [26]. Studies investigating respiratory deteriorations with enteral nutrition evaluate for aspirations or surrogates for gastric intolerance such as vomiting or gastric residual volume. We chose in our study to focus on delivered enteral nutrition volume as a marker for gastric distention, a variable well known to play a role in diaphragmatic function, as well as contribute to reflux and likely silent aspirations. [8,19,20,28]. We did not evaluate for pulmonary aspiration in our patients due to the barriers in accurately identifying this diagnosis. We instead utilized strict criteria for defining increased ventilatory support or need for ventilation as objective evidence of a worsening respiratory disease triggered by enteral nutrition delivery.

Secondly, we found that respiratory events or complications are not rare, as they occurred in 40% of the total cohort with the most common being an increase in oxygen requirement (75.9%). This change in FiO_2_ was the most commonly repeated complication in those having more than one complication. However, having more than one significant change in FiO_2_ as defined in this study, or having a pressure change on NIV was not associated with development of apnea, need for intubation, or cardiorespiratory arrest. Therefore, consideration for enteral nutrition initiation and advancement should not be limited with these changes in NIV settings.

Thirdly, our study showed that the development of more than one complication is not necessarily a predictor of a worse respiratory state and should not prevent clinicians from initiating and titrating enteral nutritional therapy. The majority of patients who developed more than one complication did not progress to develop worse complications of apnea, need for intubation, or cardiorespiratory arrest.

Fourthly, BiPAP was associated with the development of complications in this patient cohort, independent of enteral nutrition delivery. We propose two potential explanations for this finding; patients presenting with significant respiratory distress are initiated on bi-level support rather than CPAP, or the pressure delivery of BiPAP itself results in opening the lower esophageal sphincter leading to aerophagia (20–25 cmH_2_O), which is significantly associated with air ingestion, gastric distension, and respiratory complications [21]. This latter hypothesis is supported by the literature where higher NIV pressures were a significant factor associated with airway complications [9,10,19,29] and contrasts with Kogo and colleagues’ work who described no difference in rates of airway complications between BiPAP and CPAP [27].

Lastly, this study showed that the majority of our patients received enteral nutrition while on NIV as recommended by North American and European guidelines with minimal effect on rates of respiratory complications [12,13]. Multiple studies have demonstrated the large variability in EN initiation and titration between centers in critically ill adults and children [11,15,20,22,24,26]. In two recent retrospective cohort studies in children, NIV was identified as one of the factors independently associated with delayed EN initiation [10,14]. In our institution, EN is physician-prescribed and timing of initiation is determined in collaboration with dietitians and nurses managing the patients. As general practice, EN is advocated for in the first 48 h of admission. 

This study has several limitations. First, this was a single center, retrospective study and the study design limited the number of variables we could evaluate for their association with development of a complication after EN initiation. Second, we did not utilize severity of illness as a predictive variable in our patient sample. We instead focused on available bedside markers of illness severity summarized by vital signs and ventilatory settings in a patient sample with similar underlying disease pathology: acute lower respiratory tract infections. Data from other studies might suggest a variation in feeding practices according to the underlying severity of illness, whereby patients who are younger, less severely ill, and well-nourished are more likely to receive and tolerate earlier EN [14,15,30].

There were several strengths of this study. Firstly, this is one of the largest studies published on respiratory complications and enteral nutrition in patients on NIV in the pediatric population. Secondly, our study is unique in including patients on CPAP and BiPAP with exclusion of those on HHFNC that is classically used for less severe presentations of respiratory illness and is characterized by different pressure delivery into the respiratory and gastrointestinal tract. Thirdly, the included cohort is representative of the pediatric critically ill patient population with regard to age, sex, and ventilatory settings distributions allowing the generalizability of these findings. Fourthly, we predefined complications based on clinicians’ input in the ICU. The FiO_2_ and pressure cutoffs, although not previously published, are based on clinical experience where some changes in delivered O_2_ and NIV pressures would not be unusual in the care of patients with respiratory disease. Lastly, we used volume of nutrition delivered as a surrogate for gastric distention instead of focusing on route of enteral nutrition delivery or on classical signs of gastrointestinal intolerance.

## 5. Conclusions

Enteral nutrition delivery while on noninvasive ventilation is not associated with an increased risk of complications. Enteral nutrition in acute respiratory failure can improve recovery and contribute to shorter ICU stay. Prospective studies evaluating enteral nutrition in nonintubated children could attempt to quantify and validate objective clinical findings that support the clinician in the decision to initiate and titrate enteral nutrition.

## Figures and Tables

**Figure 1 nutrients-15-02817-f001:**
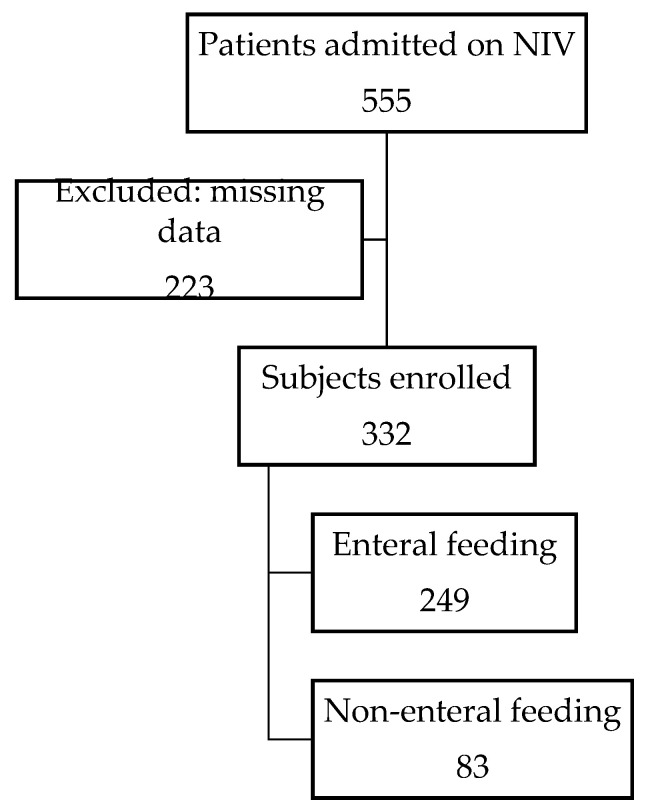
Participant flowchart. Flowchart depicting total number of admissions and number of patients who were enterally fed. NIV, noninvasive ventilation.

**Figure 2 nutrients-15-02817-f002:**
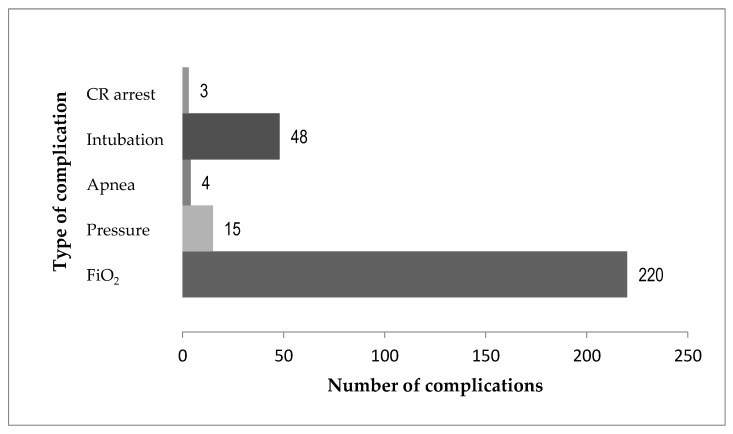
Number of complications. The total number of complications in 132 patients was 290, with the most common being an increase in oxygen requirement 220/290 (76%), followed by intubation, and then an increase in respiratory support pressure. A total of 60/132 (45%) patients had several complications. CR, cardiorespiratory; FiO_2_, fraction of inspired oxygen.

**Figure 3 nutrients-15-02817-f003:**
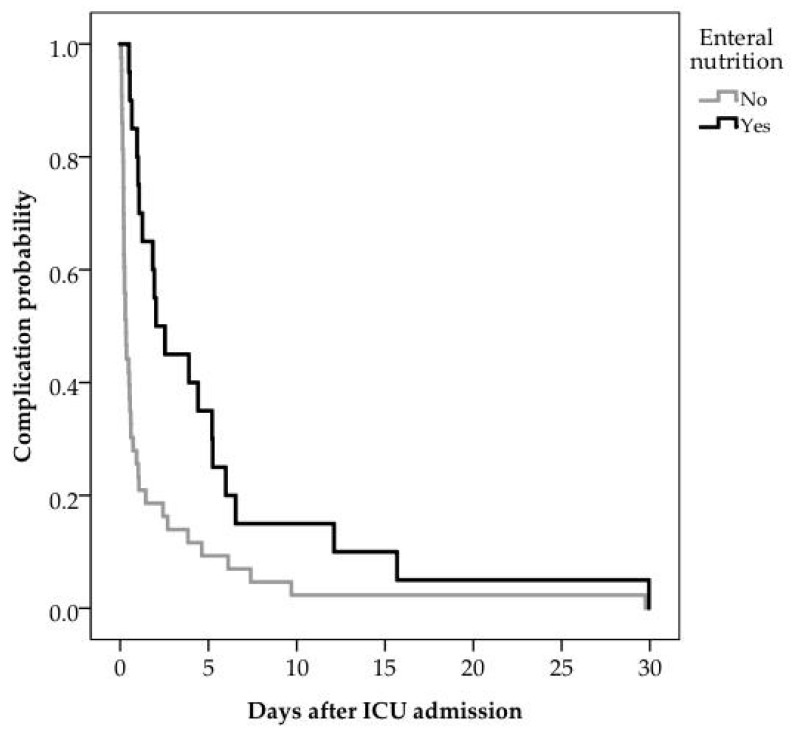
Survival curve. Days to first complication (not a change in FiO_2_). The first worst complication-free survival distributions for patients enterally fed vs. not fed were statistically significantly different (log rank 10.56; *p* < 0.01). The median complication-free survival probability in fed patients was 2.03 (95% CI 0.71–3.34), compared with 0.33 (95% CI 0.19–0.47) in not fed patients.

**Table 1 nutrients-15-02817-t001:** Significant increase in oxygen requirement.

Baseline	New FiO_2_
<30%	≥50%
31–40%	≥60%
51–60%	≥70%
61–70%	≥80%
71–80%	≥90%
>80%	100%

New FiO_2_ represents the change from baseline FiO_2_ that was considered significant for the purpose of this study. FiO_2_, fraction of inspired oxygen.

**Table 3 nutrients-15-02817-t003:** Characteristics of patients enterally fed versus not enterally fed.

Characteristics	Statistic	No Enteral Feeding	Enteral Feeding	*p*-Value
Number of patients	N (%)	83 (25)	249 (75)	
Sex, male	N (%)	48 (58)	156 (62)	0.43
Age, years	MD (Q1, Q3)	2 (0.3, 11)	2 (0.3, 8)	0.39
Weight, kg	MD (Q1, Q3)	11 (5, 27)	10 (5, 24)	0.96
NIV settings, cmH2O				
CPAP (EPAP)	MD (Q1, Q3)	7 (6, 7)	7 (6, 8)	0.36
BiPAP (EPAP)	MD (Q1, Q3)	7 (6, 8)	7 (6, 8)	0.94
BiPAP (IPAP)	MD (Q1, Q3)	14 (12, 16)	14 (12, 15)	0.13
FiO_2_	MD (Q1, Q3)	0.35 (0.3, 0.5)	0.3 (0.25, 0.45)	0.32
Vital signs				
Heart rate	MD (Q1, Q3)	130 (109, 149)	135 (113, 153)	0.40
Respiratory rate	MD (Q1, Q3)	32 (24, 38)	34 (25, 44)	0.26
SBP, mmHg	MD (Q1, Q3)	97 (89, 107)	103 (91, 114)	0.12
DBP, mmHg	MD (Q1, Q3)	55 (46, 64)	59 (49, 70)	0.08
Oxygen saturation, %	MD (Q1, Q3)	97 (95, 100)	98 (95, 99)	0.94
PICU length of stay, days	MD (Q1, Q3)	5 (1, 10)	3 (1, 6)	0.02
Hospital length of stay, days	MD (Q1, Q3)	17 (8, 40)	10 (5, 22)	0.26
Deaths	N (%)	8 (10)	13 (5)	0.15
Complication	N (%)	60 (72)	72 (29)	<0.01

Two-thirds of the cohort received enteral nutrition. There was no statistical difference in the baseline characteristics between the groups that were enterally fed or not (*p* > 0.05). Both had similar NIV settings and vital signs. However, PICU length of stay and complications were significantly higher in the non-enterally fed patients (*p* < 0.05). N, number; MD, median; Q1, first quartile; Q3, third quartile; NIV, noninvasive ventilation; CPAP, continuous positive airway pressure; BiPAP, bilevel positive airway pressure; EPAP, expiratory positive airway pressure; IPAP, inspiratory positive airway pressure; FiO_2_, fraction of inspired oxygen; SBP, systolic blood pressure; DBP, diastolic blood pressure; PICU, pediatric intensive care unit.

**Table 4 nutrients-15-02817-t004:** Vital Signs at time of Feeding Decision and Complication (N= 290).

Vital Signs	Statistic	Feeding Decision	Complication	*p*-Value
Heart rate	MD (Q1, Q3)	129 (110, 148)	132 (115, 151)	<0.01
Respiratory rate	MD (Q1, Q3)	34 (24, 45)	30 (24, 42)	0.21
SBP, mmHg	MD (Q1, Q3)	98 (86, 111)	100 (87, 112)	0.07
DBP, mmHg	MD (Q1, Q3)	55 (46, 67)	58 (48, 69)	0.20
Oxygen saturation, %	MD (Q1, Q3)	97 (93, 99)	94 (88, 98)	<0.01

Patients’ heart rate was significantly higher and saturations were significantly lower around the time of developing a complication in comparison to the time of initiation of feeds (*p* < 0.01). MD, median; Q1, first quartile; Q3, third quartile; SBP, systolic blood pressure; DBP, diastolic blood pressure.

**Table 5 nutrients-15-02817-t005:** Characteristics of patients who developed a complication.

Characteristics	Statistic	No Enteral Feeding	Enteral Feeding	*p*-Value
Number of patients	N	60	72	
Sex, male	N (%)	33 (55)	51 (71)	0.06
Age, years	MD (Q1, Q3)	1 (0.2, 11)	2 (0.4, 8)	0.71
Weight, kg	MD (Q1, Q3)	8 (5, 25)	10 (5, 22)	0.76
NIV settings, cmH2O				
CPAP	MD (Q1, Q3)	7 (6, 7)	6 (5, 8)	0.65
BiPAP (EPAP)	MD (Q1, Q3)	7 (6, 8)	7 (6, 8)	0.35
BiPAP (IPAP)	MD (Q1, Q3)	14 (12, 16)	14 (12, 16)	0.47
FiO_2_	MD (Q1, Q3)	0.35 (0.3, 0.5)	0.36 (0.3, 0.53)	0.63
Vital signs				
Heart rate	MD (Q1, Q3)	130 (109, 149)	136 (114, 152)	0.33
Respiratory rate	MD (Q1, Q3)	32 (24, 38)	34 (25, 47)	0.17
SBP, mmHg	MD (Q1, Q3)	97 (89, 107)	97 (86, 112)	0.89
DBP, mmHg	MD (Q1, Q3)	55 (46, 64)	56 (46, 67)	0.83
Oxygen saturation, %	MD (Q1, Q3)	97 (95, 99)	96 (93, 99)	0.19
PICU length of stay, days	MD (Q1, Q3)	8 (5, 14)	6 (3, 12)	0.21
Hospital length of stay, days	MD (Q1, Q3)	21 (10, 44)	15 (8, 39)	0.99
Deaths	N (%)	6 (10)	8 (11)	0.84

Patients who developed complications with and without enteral feeding had similar NIV settings and vital signs around the time of complication. N, number; MD, median; Q1, first quartile; Q3, third quartile; NIV, noninvasive ventilation; CPAP, continuous positive airway pressure; BiPAP, bilevel positive airway pressure; EPAP, expiratory positive airway pressure; IPAP, inspiratory positive airway pressure; FiO_2_, fraction of inspired oxygen; SBP, systolic blood pressure; DBP, diastolic blood pressure; PICU, pediatric intensive care unit.

**Table 6 nutrients-15-02817-t006:** Factors associated with the development of the first complication.

Characteristics	STAT	Complication	No Complication	Univariate	Multivariate	
				*p*	OR	*p*
Number of patients	N	132	200			
Sex, male	N (%)	84 (63.6)	120 (60)	0.50		
Age, years	M (±SD)	4.82 (±5.73)	4.86 (±5.48)	0.95		
Weight, kg	M (±SD)	17.7 (±18.6)	18.0 (±17.9)	0.86		
NIV, BiPAP	N (%)	23 (17.4)	96 (48)	<0.01	5.33	<0.01
NIV settings, cmH2O	N (%)	109 (82.6)	104 (52)			
CPAP (EPAP)	M (±SD)	7 (±2)	7 (±1)	0.26		
BiPAP (EPAP)	M (±SD)	7 (±1)	7 (±1)	0.22		
BiPAP (IPAP)	M (±SD)	14 (±2)	13 (±2)	0.12		
FiO_2_	M (±SD)	0.42 (±0.18)	0.35 (±0.17)	<0.01	6.00	0.03
No enteral feeds	N (%)	60 (45)	23 (11.5)	<0.01	0.01	1.00
Feeds rate, mL/kg						
1 h	M (±SD)	3.84 (±4.78)	6.81 (±5.87)	<0.01	0.90	0.21
4 h	M (±SD)	9.95 (±9.25)	15.12 (±12.8)	<0.01	0.95	0.16
Vital signs						
Heart rate	M (±SD)	132 (±29)	133 (±30)	0.91		
Respiratory rate	M (±SD)	36 (±16)	35 (±33)	0.90		
SBP, mmHg	M (±SD)	98 (±16)	104 (±17)	<0.01	0.98	0.05
DBP, mmHg	M (±SD)	57 (±15)	62 (±16)	<0.01	0.99	0.39
Oxygen saturation, %	M (±SD)	95 (±5)	97 (±5)	<0.01	0.96	0.22

In univariate analysis, the factors being on BiPAP, having higher FiO_2_, and not being fed (or being fed at lower rates) were associated with developing a complication. In the multivariate model, the factors associated with the occurrence of a complication were need for BiPAP and higher FiO_2_ requirements. Feeding status did not reach statistical significance in the multivariate assessment. N, number; M, mean; SD, standard deviation; STAT, statistic; NIV, noninvasive ventilation; CPAP, continuous positive airway pressure; BiPAP, bilevel positive airway pressure; EPAP, expiratory positive airway pressure; IPAP, inspiratory positive airway pressure; FiO_2_, fraction of inspired oxygen; SBP, systolic blood pressure; DBP, diastolic blood pressure; PICU, pediatric intensive care unit.

**Table 7 nutrients-15-02817-t007:** Enteral feeds and vital signs association with the development of a complication.

Characteristics	STAT	Complication	No Complication	Univariate	Multivariate	
				*p*	OR	*p*
Number of events	N	290	200			
No enteral feeds	N (%)	104 (36)	23 (11.5)	<0.01	3.56	<0.01
Feeds rate, mL/kg						
1 h	M (±SD)	3.30 (±4.91)	6.03 (±5.94)	<0.01	0.92	0.12
4 h	M (±SD)	8.36 (±11)	13.39 (±13)	<0.01	1.01	0.79
Vital signs						
Heart rate	M (±SD)	132 (±27)	126 (±23)	0.01	1.01	0.12
Respiratory rate	M (±SD)	35 (±15)	34 (±11)	0.6		
SBP, mmHg	M (±SD)	100 (±17)	102 (±14)	0.09	1	0.74
DBP, mmHg	M (±SD)	59 (±15)	59 (±10)	0.59		
Oxygen saturation, %	M (±SD)	91 (±13)	97 (±4)	<0.01	0.82	<0.01

In univariate analysis the factors associated with the occurrence of a complication were lack of enteral nutrition, low enteral nutrition rates, heart rate, and oxygen saturation. There were no differences in the respiratory rate or blood pressure between the groups. A total of 49/290 complications had incomplete respiratory support data. In the multivariate model, the factors not being fed and having lower oxygen saturation were at higher risk of developing a complication. N, number; M, mean; SD, standard deviation; STAT, statistic; SBP, systolic blood pressure; DBP, diastolic blood pressure.

## Data Availability

The data presented in this study are available on request from the corresponding author.

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
