# Peer review of "A Retrospective Study of Complications of Enteral Feeding in Critically Ill Children on Noninvasive Ventilation"

_nutrients, 2023, doi:10.3390/nu15122817_

Round 1
Reviewer 1 Report
In their work, the authors presented a retrospective study in which they examined the association between enteral nutrition and the development of extraintestinal complications in children aged 0 to 18 years on NIV for acute respiratory failure. Overall the work is well written.
Comments:
Abstract: All results mentioned require numbers (%, N and p values). The abbreviation BiPAP must be explained in whole words. Also, check others in the text.
Results: All tables must include in the first raw data easier to follow: e.g., N (%); mean (SD); N(median) or similar (not in the legend)
References: need updating.
English needs grammar checking.
Reviewer 2 Report
The authors by means of a retrospective study, evaluated the safety of enteral feeds in children requiring NIV acutely. In this study a representative sample of the pediatric critically ill patient population in regards to age, sex, and ventilatory settings distributions was analysed, demonstrating that enteral nutrition in acute settings of respiratory failure can improve recovery and contribute to shorter intensive care unit stay.
In this paper clarity of presented data could be improved and the authors should adress the following concerns:
1- Please include in the abstract and in the main text what all the abbreviations stand for: e.g. BiPAP (line 25), ICU (line 28), HR (line 143), BP Line 144) . It could be useful to add a complete list of abbreviations used, as a new paragraph.
2- Please check and fix FiO2 troughtout the text using the subscript correctly.
3- The lines from 54 to 61 represent a point for the discussion rather than a part of the introduction.
4- Tables 2, 4 and 6 require graphics review, same lines are missing.
5- The number and the description of the tables can be put together, on the same line
The text need revision to improve comprehension and fluency
Round 2
Reviewer 1 Report
The authors havel addressed all issues.